

# The parastatistics of braided Majorana fermions

**Francesco Toppan**⋆

Centro Brasileiro de Pesquisas Físicas CBPF, Rua Dr. Xavier Sigaud 150,
Urca, cep 22290-180, Rio de Janeiro (RJ), Brazil.

⋆ toppan@cbpf.br

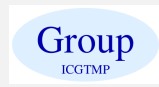

## Abstract

This paper presents the parastatistics of braided Majorana fermions obtained in the framework of a graded Hopf algebra endowed with a braided tensor product. The braiding property is encoded in a $t$-dependent $4 \times 4$ braiding matrix $B_t$ related to the Alexander-Conway polynomial. The nonvanishing complex parameter $t$ defines the braided parastatistics. At $t = 1$ ordinary fermions are recovered. The values of $t$ at roots of unity are organized into levels which specify the maximal number of braided Majorana fermions in a multiparticle sector. Generic values of $t$ and the $t = -1$ root of unity mimick the behaviour of ordinary bosons.

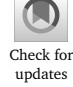

## 1 Introduction

Braided Majorana fermions have been intensively investigated since the [1] Kitaev's proposal that they can be used to encode the logical operations of a topological quantum computer which offers protection from decoherence (see also [2–4]). In this talk I present consequences and open questions about the parastatistics of $\mathbb{Z}_2$-graded braided Majorana qubits derived from the results of [5]; this paper applied to $\mathbb{Z}_2$-graded qubits the [6] framework of a graded Hopf algebra endowed with a braided tensor product. A nonvanishing complex braiding parameter $t$ controls the spectra of multiparticle Majorana fermions. Inequivalent physics is derived for the set of $t$ roots of unity which are organized into different levels $(L_2, L_3, \dots, L_\infty)$. The levels interpolate between ordinary fermions ($L_2$ for $t = 1$) and the spectrum of bosons ("$L_\infty$" recovered at $t = -1$). The intermediate levels $L_k$ for $k = 3, 4, 5, \dots$ implement a special type of parafermionic statistics (see [7–9]) which allows at most $k - 1$ braided Majorana excited states in any given multiparticle sector.

The paper is structured as follows. In Section 2 the braiding of $\mathbb{Z}_2$-graded qubits is illustrated. In Section 3 the truncations of the spectra at roots of unity are discussed. The consequences for the parastatistics are presented in Section 4.

## 2 Braiding $\mathbb{Z}_2$-graded qubits

We present the main ingredients of the construction. A single Majorana fermion can be described as a $\mathbb{Z}_2$-graded qubit which defines a bosonic vacuum state $|0\rangle$ and a fermionic excited state $|1\rangle$:

$$|0\rangle = \begin{pmatrix} 1 \\ 0 \end{pmatrix}, \qquad |1\rangle = \begin{pmatrix} 0 \\ 1 \end{pmatrix}. \tag{1}$$

The operators acting on the $\mathbb{Z}_2$-graded qubit close the $\mathfrak{gl}(1|1)$ superalgebra. In a convenient presentation they can be defined as

$$\alpha = \begin{pmatrix} 1 & 0 \\ 0 & 0 \end{pmatrix}, \quad \beta = \begin{pmatrix} 0 & 1 \\ 0 & 0 \end{pmatrix}, \quad \gamma = \begin{pmatrix} 0 & 0 \\ 1 & 0 \end{pmatrix}, \quad \delta = \begin{pmatrix} 0 & 0 \\ 0 & 1 \end{pmatrix}. \tag{2}$$

Their (anti)commutators are

$$[\alpha,\beta] = \beta, \qquad [\alpha,\gamma] = -\gamma, \qquad [\alpha,\delta] = 0, \qquad [\delta,\beta] = -\beta, \qquad [\delta,\gamma] = \gamma,$$
$$\{\beta,\beta\} = \{\gamma,\gamma\} = 0, \qquad \{\beta,\gamma\} = \alpha + \delta. \tag{3}$$

The diagonal operators $\alpha, \beta$ are even, while $\beta, \gamma$ are odd, with $\gamma$ the fermionic creation operator.

The excited state is a Majorana since it is a fermion which coincides with its own antiparticle. This is a consequence of the fact that the (2) matrices span the Clifford algebra $Cl(2,1)$ which, see [10, 11], is of real type (implying that the charge conjugation operator is the identity).

The construction of multiparticle $\mathbb{Z}_2$-graded qubits is obtained via the coproduct $\Delta$ of the graded Hopf algebra $\mathcal{U}(\mathfrak{gl}(1|1))$, the Universal Enveloping Algebra of $\mathfrak{gl}(1|1)$.

The braiding of the graded qubits is realized by introducing a braided tensor product $\otimes_{br}$ such that, for the operators $a, b$ ($\mathbb{I}$ is the identity) one can write

$$(\mathbb{I} \otimes_{br} a) \cdot (b \otimes_{br} \mathbb{I}) = \Psi(a,b), \tag{4}$$

where the right hand side operator $\Psi(a,b)$ satisfies braided compatibility conditions.

For the purpose of braiding $\mathbb{Z}_2$-graded qubits it is only necessary to specify the braiding property of the creation operator $\gamma$:

$$(\mathbb{I} \otimes_{br} \gamma) \cdot (\gamma \otimes_{br} \mathbb{I}) = \Psi(\gamma,\gamma). \tag{5}$$

A consistent choice for the right hand side is to set

$$\Psi(\gamma,\gamma) = B_t \cdot (\gamma \otimes \gamma), \tag{6}$$

where $B_t$ is a $4 \times 4$ constant matrix which depends on the complex parameter $t \neq 0$. The dot in the right hand side denotes the standard matrix multiplication.

The braiding compatibility condition is guaranteed by assuming $B_t$ to be given by

$$B_t = \begin{pmatrix} 1 & 0 & 0 & 0 \\ 0 & 1-t & t & 0 \\ 0 & 1 & 0 & 0 \\ 0 & 0 & 0 & -t \end{pmatrix}, \tag{7}$$

since $B_t$ satisfies

$$(B_t \otimes \mathbb{I}_2) \cdot (\mathbb{I}_2 \otimes B_t) \cdot (B_t \otimes \mathbb{I}_2) = (\mathbb{I}_2 \otimes B_t) \cdot (B_t \otimes \mathbb{I}_2) \cdot (\mathbb{I}_2 \otimes B_t). \tag{8}$$

The matrix $B_t$ is the $R$-matrix of the Alexander-Conway polynomial in the linear crystal rep on exterior algebra [12] and is related, see [13], to the Burau representation of the braid group.

## 3 Truncations at roots of unity

The requirement that

$$B_t^n = \mathbb{I}_4 \,, \tag{9}$$

for some $n = 2, 3, \dots$ finds solution for the $n-1$ roots of the polynomial $b_n(t)$. This set of polynomials is defined as

$$b_{n+1}(t) = \sum_{j=0}^{n} (-t)^j \,,$$

so that

$$
\begin{aligned}
b_1(t) &= 1 \,, \\
b_2(t) &= 1 - t \,, \\
b_3(t) &= 1 - t + t^2 \,, \\
b_4(t) &= 1 - t + t^2 - t^3 \,, \\
b_5(t) &= 1 - t + t^2 - t^3 + t^4 \,, \\
\dots &= \dots
\end{aligned}
$$

The set of $b_k(t)$ polynomials enters the construction of multiparticle states. The $n$-particle vacuum $|0\rangle_n$ is given by the tensor product of $n$ single-particle vacua:

$$|0\rangle_n = |0\rangle \otimes |0\rangle \otimes \dots \otimes |0\rangle \qquad (n \text{ times}) \,. \tag{10}$$

The fermionic excited states are created by applying powers of tensor products involving the single-particle creation operator $\gamma$. For $n = 2, 3$ one has, e.g., that the first excited state is created by

$$
\begin{aligned}
\gamma_{(2)} &= \mathbb{I}_2 \otimes_{br} \gamma + \gamma \otimes_{br} \mathbb{I}_2 \,, \\
\gamma_{(3)} &= \mathbb{I}_2 \otimes_{br} \mathbb{I}_2 \otimes_{br} \gamma + \mathbb{I}_2 \otimes_{br} \gamma \otimes_{br} \mathbb{I}_2 + \gamma \otimes_{br} \mathbb{I}_2 \otimes_{br} \mathbb{I}_2 \,. 
\end{aligned} \tag{11}
$$

By taking into account the braided tensor product one obtains, for the second and third excited states,

$$
\begin{aligned}
\gamma_{(2)}^2 &= (1-t) \cdot (\gamma \otimes_{br} \gamma) \,, \\
\gamma_{(3)}^2 &= (1-t) \cdot (\mathbb{I}_2 \otimes_{br} \gamma \otimes_{br} \gamma + \gamma \otimes_{br} \mathbb{I}_2 \otimes_{br} \gamma + \gamma \otimes_{br} \gamma \otimes_{br} \mathbb{I}_2) \,, \\
\gamma_{(3)}^3 &= (1-t)(1-t+t^2) \cdot (\gamma \otimes_{br} \gamma \otimes_{br} \gamma) \,.
\end{aligned}
$$

This construction works in general. The $b_k(t) = 0$ roots of the polynomials produce truncations at the higher order excited states and the corresponding spectrum of the theory.

## 4 The levels and the associated parastatistics

The single-particle Hamiltonian $H$ can be identified with the operator $\delta$ in (2). It follows that the single-particle excited state has energy level $E = 1$. This is also true (due to the property of the Hopf algebra coproduct) for the first excited state in the multiparticle sector. Each creation operator produces a quantum of energy.

In the $n$-particle sector the energy spectrum of the theory depends on whether $t$ produces a truncated or untruncated spectrum. The notion of truncation level acquires importance.

A "level-$k$" root of unity, for $k = 2, 3, 4, \ldots$, is a a solution $t_k$ of the $b_k(t_k) = 0$ equation such that, for any $k' < k$, $b_{k'}(t_k) \neq 0$.

The physical significance of a level-$k$ root of unity is that the corresponding braided multiparticle Hilbert space can accommodate at most $k - 1$ Majorana spinors.

The special point $t = 1$, being the solution of the $b_2(t) \equiv 1 - t = 0$ equation, is a level-2 root of unity. It gives the ordinary total antisymmetrization of the fermionic wavefunctions. The $t = 1$ level-2 root of unity encodes the Pauli exclusion principle of ordinary fermions.

With an abuse of language, the $t = -1$ root of unity, which does not solve any $b_k(t) = 0$ equation, can be called a root of unity of $\infty$ level.

The physics does not depend on the specific value of $t$, but only on the root of unity level. A generic $t$ which does not coincide with a root of unity produces the same untruncated spectrum of the $t = -1$ "$L_\infty$" level.

The following energy spectra are derived.

**Case a, truncated $L_k$ level:** the $n$-particle energy eigenvalues $E$ are

$$E = 0, 1, \ldots, n, \qquad \text{for} \quad n < k,$$
$$E = 0, 1, \ldots, k - 1, \qquad \text{for} \quad n \geq k;$$

a plateau is reached for the maximal energy level $k - 1$; this is the maximal number of braided Majorana fermions that can be accommodated in a multiparticle Hilbert space;

**Case b, untruncated ($t = -1$) $L_\infty$ level:** the $n$-particle energy eigenvalues $E$ are

$$E = 0, 1, \ldots, n, \qquad \text{for any} \quad n;$$

there is no plateau in this case. The energy eigenvalues grow linearly with $N$.

We can associate the roots of unity levels to fractions.

Let $t = e^{i\theta} = e^{if\pi}$ with $f \in [0, 2[$. The following fractions correspond to the roots of unity levels:

$$
\begin{aligned}
L_\infty &= 1, \\
L_2 &= 0, \\
L_3 &= \frac{1}{3}, \frac{5}{3}, \\
L_4 &= \frac{1}{2}, \frac{3}{2}, \\
L_5 &= \frac{1}{5}, \frac{3}{5}, \frac{7}{5}, \frac{9}{5}, \\
L_6 &= \frac{2}{3}, \frac{4}{3}, \\
L_7 &= \frac{1}{7}, \frac{3}{7}, \frac{5}{7}, \frac{9}{7}, \frac{11}{7}, \frac{13}{7}, \\
L_8 &= \frac{1}{4}, \frac{3}{4}, \frac{5}{4}, \frac{7}{4}, \\
\ldots &= \ldots
\end{aligned}
$$

As an example, the 5 roots of $b_6(t) = 1 - t + t^2 - t^3 + t^4 - t^5$ are classified, for $t = exp(i\theta)$, into:

level-2 root, $\theta = 0$,
level-3 roots $\theta = \pi/3$ and $5\pi/3$,
level-6 roots $\theta = 2\pi/3$ and $4\pi/3$.

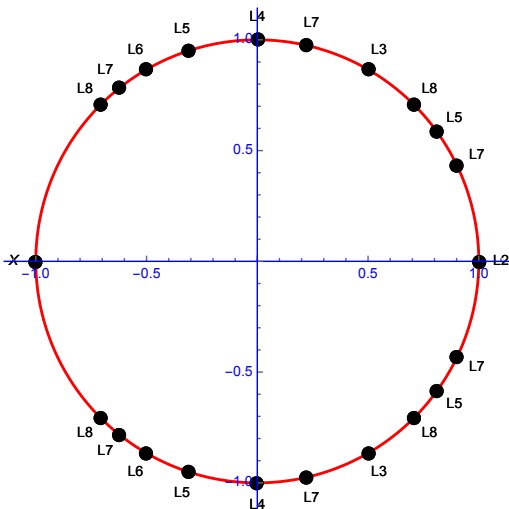

Figure 1: Roots of unity up to level 8.

The above figure shows how the roots of unity are accommodated up to level 8.

The level $k$ root accommodates at most $k$ inequivalent energy levels in the multiparticle states.

## 5 Conclusion

The [5] braided multiparticle quantization of Majorana fermions produces truncations of the spectra at certain values of $t$ roots of unity. This feature points towards a relation between the braided tensor product framework here discussed and the representations of quantum groups at roots of unity where similar truncations, see [14, 15], are observed. The precise connection of the two approaches is on the other hand not yet known and still an open question. The representations of the quantum group $\mathcal{U}_q(\mathfrak{gl}(1|1))$ at roots of unity have been classified and presented in [16] (see also [17]). A possibility to investigate the connection seems to be offered by the scheme of [18] which shows how a quasitriangular Hopf algebra can be converted into a braided group.

On a separate issue it should be mentioned that a forthcoming paper will present, with the help of intertwining operators, the construction of the braided tensor product $\otimes_{br}$ in terms of an ordinary tensor product $\otimes$. This construction relates the observed parastatistics of Majorana fermions to the "mixed brackets" (which interpolate ordinary commutators and anticommutators) that have been introduced in [19] in defining the Volichenko algebras.

## Acknowledgements

**Funding information** This work was supported by CNPq (PQ grant 308846/2021-4).

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
