# Peer review of "The parastatistics of braided Majorana fermions"

_SciPost Physics Proceedings, doi:SciPost Phys. Proc. 14, 046 (2023)_

## Round 1 · Referee Report · Anonymous (Referee 1) · 2022-12-28

Weaknesses
Unclear if this has any physics applications. Much is repeated from earlier work by same author.
Report
I have read "The parastatistics of braided Majorana fermions" by
Francesco Toppan. Toppan has constructed a representation of the
braid group based on the burau representation. One can debate the
meaning of this result (and whether or not the use of the word
"Majorana" is actually appropriate here). I will not argue these
points. My main complaint about this work is simply that it is not
going to be of much interest to most of the readers of SciPost. To
start with, much of the work is just a summary of an earlier work (Nuc
Phys B 980) by the same author and very little further is new.
Secondly, there is no attempt to identify any reasonable physical
system that this mathematics applies to. I therefore do not think
this work is publishable in SciPost, but rather should be submitted to
a specialist journal.
Francesco Toppan. Toppan has constructed a representation of the
braid group based on the burau representation. One can debate the
meaning of this result (and whether or not the use of the word
"Majorana" is actually appropriate here). I will not argue these
points. My main complaint about this work is simply that it is not
going to be of much interest to most of the readers of SciPost. To
start with, much of the work is just a summary of an earlier work (Nuc
Phys B 980) by the same author and very little further is new.
Secondly, there is no attempt to identify any reasonable physical
system that this mathematics applies to. I therefore do not think
this work is publishable in SciPost, but rather should be submitted to
a specialist journal.

Author: Francesco Toppan on 2023-02-06 [id 3314]
(in reply to Report 1 on 2022-12-28)Since the SciPost web interface does not allow me to submit the revision requested by the Editor
I use this window to attach the pdf file of my revision where I point out why the fermions considered
in the paper are Majorana's.
Attachment:
gr34toppanrevised.pdf

---

## Round 2 · Author Response

Dear Editors,
following your request of a minor revision I have inserted a paragraph
explaining why the Z_2-graded qubits describe Majorana fermions.
Two extra references ([10] and [11]) have been added with respect
to the previous version.
Sincerely Yours,
Francesco Toppan
following your request of a minor revision I have inserted a paragraph
explaining why the Z_2-graded qubits describe Majorana fermions.
Two extra references ([10] and [11]) have been added with respect
to the previous version.
Sincerely Yours,
Francesco Toppan

---

## Round 2 · List of Changes

A paragraph (4 lines) has been added after formula (3) at page 2:
from “The excited state is a Majorana …”
till “ … (implying that the charge conjugation operator is the identity).”
Two extra references ([10] and [11]) have been added.
from “The excited state is a Majorana …”
till “ … (implying that the charge conjugation operator is the identity).”
Two extra references ([10] and [11]) have been added.

---

## Editorial Decision

published